# Novel use of XSTAT 30 for mitigation of lethal non-compressible torso hemorrhage in swine

**Alicia M. Bonanno**[1], **Todd L. Graham**[2], **Lauren N. Wilson**[2], **James D. Ross**[2,3¤]*

**1** Department of Surgery, Oregon Health and Science University, Portland, Oregon, United States of America, **2** Division of Trauma and Acute Care Surgery, Oregon Health and Science University, Portland, Oregon, United States of America, **3** Charles T. Dotter Department of Interventional Radiology, Oregon Health and Science University, Portland, Oregon, United States of America

¤ Current address: Military & Health Research Foundation, Laurel, Maryland, United States of America
* rosja@ohsu.edu

**Data Availability Statement:** All relevant data are within the manuscript and its Supporting Information files.

## Abstract

### Background

Management of Non-Compressible Torso Hemorrhage (NCTH) consists primarily of aortic occlusion which has significant adverse outcomes, including ischemia-reperfusion injury, in prolonged field care paradigms. One promising avenue for treatment is through use of RevMedx XSTAT 30™ (an FDA approved sponge-based dressing utilized for extremity wounds). We hypothesized that XSTAT 30™ would effectively mitigate NCTH during a prolonged pre-hospital period with correctable metabolic and physiologic derangements.

### Methods and findings

Twenty-four male swine (53±2kg) were anesthetized, underwent line placement, and splenectomy. Animals then underwent laparoscopic transection of 70% of the left lobe of the liver with hemorrhage for a period of 10min. They were randomized into three groups: No intevention (CON), XSTAT 30™-Free Pellets (FP), and XSTAT 30™-Bagged Pellets (BP). Animals were observed for a pre-hospital period of 180min. At 180min, animals underwent damage control surgery (DCS), balanced blood product resuscitation and removal of pellets followed by an ICU period of 5 hours. Postoperative fluoroscopy was performed to identify remaining pellets or bags. Baseline physiologic and injury characteristics were similar. Survival rates were significantly higher in FP and BP (p<0.01) vs CON. DCS was significantly longer in FP in comparison to BP (p = 0.001). Two animals in the FP group had pellets discovered on fluoroscopy following DCS. There was no significant difference in blood product or pressor requirements between groups. End-ICU lactates trended to baseline in both FP and BP groups.

### Conclusions

While these results are promising, further study will be required to better understand the role for XSTAT in the management of NCTH.

**Funding:** This work was supported by Military Health Research Foundation, Award Number: Contract SC-15-03, PO 0006. The funders had no role in study design, data collection and analysis, decision to publish, or preparation of the manuscript.

**Competing interests:** The authors have declared that no competing interests exist.

## Introduction

Traumatic non-compressible torso hemorrhage (NCTH) is the most common cause of potentially survivable deaths in both civilian and military trauma [1, 2]. Current treatment for junctional and torso hemorrhage consist of balanced blood product resuscitation, resuscitative thoracotomy with aortic cross clamping and placement of endovascular balloon occlusion of the aorta when logistically feasible [3, 4]. However, in austere environments, there may be a decreased availability of blood products. Additionally, increasing transport times limit the usefulness of extended Zone 1 aortic occlusion due to the high risk of ischemia reperfusion injury [5, 6]. Hemostatic gauzes, such as QuikClot Combat Gauze, have been used to manage junctional and extremity hemorrhage, however are not compatible with injury patterns that include abdominal hemorrhage [7].

XSTAT 30™, an FDA-approved sponge based hemostatic dressing that is used for the management of extremity bleeding, has also shown promise in the control of non-compressible junctional hemorrhage [8]. The device is composed of an applicator filled with highly compressed sponges that expand in the presence of a liquid. The individual compressed sponges (pellets) that make up the XSTAT 30™ have radio-opaque markers for identification and have most recently been refined for use in severe uterine hemorrhage where sponges are deployed within a radio-opaque bag [9]. Through these developments, XSTAT 30™ may have a broader application for pre-hospital intra-abdominal hemorrhage control.

The purpose of this study was to determine the efficacy of XSTAT 30™ in pre-hospital management of NCTH in a swine model. We hypothesized the following: $H1_0$ XSTAT 30™ would not elicit a survival benefit in an animal model of NCTH compared to Hextend resuscitation alone as evidenced by pre-hospital survival and intra-cavitary shed blood; $H2_0$ physiologic derangements incurred following pre-hospital application of XSTAT 30™ would not be amenable to correction by modern damage control therapies as evidence by significant disturbances in metabolic parameters to include but not limited to: serum lactate, pH, base excess and bicarbonate at the end of intensive care observation.

## Materials and methods

Oregon Health and Science University's Institutional Animal Care and Use Committee reviewed and approved this protocol prior to the initiation of the study. Telazol 8mg/kg IM was used for sedation. Inhaled Isoflurane was used for anesthesia. Euthanasia solution 1mL/10lb was used for euthanasia. Experiments were carried out in a facility accredited by the Association for Assessment and Accreditation of Laboratory Animal Care at Oregon Health and Science University, Portland, Oregon. Animals were used in accordance with *The Guide for the Care and Use of Laboratory Animals*. A priori power analysis was performed which demonstrated an estimated 60% survival difference detected with n = 8 animals.

Twenty-four male Yorkshire swine (53±2kg) from a single source vendor (Oak Hill Genetics Inc, Ewing, IL, USA) were anesthetized with Telazol (Zoetis, Persippany-Troy Hills, NJ, USA) IM injection and maintained with Isoflurane after oro-tracheal intubation for the entirety of the experiment until euthanasia. No animals regained consciousness after initial sedation. Following intubation, the animals were given buprenorphine (IM) injection for intra-operative analgesia. Vascular access was achieved by Seldinger technique and included right external jugular pulmonary artery catheter, left carotid artery catheter, two left external jugular venous catheters, and femoral arterial and venous catheters. Cut-down was performed for right carotid artery access for carotid flow probe monitor placement. Physiologic telemetry was monitored, including the following: EKG, invasive arterial pressures, cardiac output,

systemic vascular resistance, $EtCO_2$, $SvO_2$, $VO_2$, tissue oxygenation by near-infrared spectroscopy ($StO_2$), and carotid flow.

The investigators implemented a previously described swine model of NCTH as follows [3, 10]: Laparotomy and devascularization of the spleen (which was left in the anatomical position) were performed to eliminate auto-transfusion and create a standardized degree of soft tissue injury. Laparoscopic ports were placed to include two 12mm ports in the right lower quadrant, one 5mm port in the left lower quadrant and one 12mm port in the left lower quadrant. An intra-abdominal balloon transducer (Cook Quantum TTC Colonic Balloon Dilator 6FR, 5.5cm x 1.6cm) was placed into the left subdiaphragmatic area for intra-abdominal pressure monitoring. The abdomen was then closed with a bespoke incorporated 30mm trocar for XSTAT 30™ deployment at upper midline of the incision and the animal underwent a ten-minute stabilization period. Blood samples were taken following stabilization to confirm no significant physiologic changes occurred.

Following stabilization, the abdomen was insufflated to 15mmHg and laparoscopic liver injury was created in all animals by transecting approximately 70% of the left lateral lobe of the liver. The abdomen was desufflated, all ports were removed with the exception of the XSTAT 30™ trocar, and the skin was reapproximated using staples. Free bleeding was allowed to occur for a period of 10 minutes during which animals were randomized one of three groups: Control–No intervention (CON), XSTAT 30™ with free pellets (FP), XSTAT 30™ with pellets in radiopaque bags (BP). This control was specifically chosen as this model has previously been demonstrated to have lethal effects with no intervention [10]. Randomization schedule was achieved by selecting an unlabeled, sealed envelope that contained one of the three experimental groups and had been prepared by a third-party individual.

All animals underwent infusion of up to 2 boluses of 300mL of Hextend at initial intervention if MAP<50 and SBP<90mmHg. For animals randomized to the FP group, six XSTAT 30™ applicators were deployed through the central 30mm trocar to replicate a penetrating injury while still being able to complete the laparoscopic transection. In the BP group, six XSTAT 30™ devices were deployed within multiple radiopaque bags (33–34 pellets/bag, 3 bags/applicator for a total of 100 pellets/applicator). The total number of pellets administered was the same (600 pellets) regardless of experimental group. The XSTAT 30™ pellets whether in bags or free were deployed per manufacturers guidelines. Animals in all groups were then observed for 180 minutes in a simulated pre-hospital phase.

Following the pre-hospital phase, the animals underwent a standardized damage control surgery (DCS) for primary control of hemorrhage and received up to 3L of whole blood resuscitation to simulate a balanced blood product resuscitation. Upon entering the abdomen, the pellets or bags overlying the liver injury were removed until manual control of the liver could be achieved. Once definitive hemorrhage control was determined by placing atraumatic liver clamps on the transected edge of the liver, the remaining pellets and bags within adjacent areas were collected. Pellets deployed in bags were counted to ensure complete removal and the entire abdomen was inspected for any remaining pellets. Shed blood was collected following removal of pellets. The liver was then packed to control for any additional bleeding. The abdomen was closed with a Bogota bag made from a 1L bag of NS and the animal was observed for an additional 300 minutes of ICU time [11]. During the ICU phase of the experiment, a critical care algorithm was utilized to maintain a normotensive state and to correct any electrolyte abnormalities. If the animal's SBP<90mmHg, and Hgb≥7g/dL they received fresh frozen plasma (FFP). If the animal's Hgb<7g/dL they received pRBCs. If the animal's pressures were unresponsive to two subsequent boluses either FFP or pRBCs and their SVR<80% of baseline, norepinephrine was initiated at 0.02 mcg/kg/min. Insulin and 50mL of 50% Dextrose was

administered for potassium≥5.5mmol/L. For glucose<3.0mmol/L, 50mL of 50% Dextrose was administered.

Subsequent DCS, the animals underwent fluoroscopy of the entire abdomen to identify any remaining pellets that were not collected at the time of surgery. If pellets or bags were identified by radiologic imaging, they were noted and retrieved during necropsy.

At the completion of the experiment, animals were euthanized with 1mL/10lb Euthanasia Solution (MWI Animal Health, Boise, ID, USA) and underwent necropsy and gross pathological assessment. Tissue samples were collected from the terminal ileum, kidney, liver, pancreas, lung and heart. These samples were sent for histological analysis by a certified veterinary pathologist.

## Data collection and study endpoints

Arterial blood samples were taken pre-surgery (baseline), end of prep and stabilization (T = 0), intervention (T = 10), T = 30, T = 45 and then time intervals of 30 minutes beginning at T = 60 until end of experiment (T = 480) or at the time of euthanasia. Data collected for the $H1_0$ primary endpoint included survival at end of pre-hospital and end of study. Secondary endpoints for $H1_0$ included indices of cardiovascular and cardiopulmonary function including recorded measurements of cardiac output, intra-abdominal pressure changes, shed blood and remaining pellets within the abdomen. Data collected for the $H2_0$ primary endpoint were serum lactate. Secondary endpoints included recorded indices of tissue oxygenation and oxygen consumption, blood gas and chemistry, clotting and coagulation status, and histopathological examination of lung, apex of the heart, kidney, liver, pancreas and bowel for evidence of ischemic injury.

## Statistical analysis

Baseline and end pre-hospital data points were compared using ANOVA. End-study data points were compared using Student's t-test. Categorical data was analyzed with Fischer's exact test. Clinical and laboratory values were measured, and analyzed by ANOVA with post-hoc pairwise comparisons where appropriate (Holm-Sidak). Kaplan-Meier analysis was performed for survival. All statistical analysis was completed using SigmaPlot 12 (SyStat Software Inc.).

## Results

### Baseline animal characteristics

There were no statistical differences in baseline characteristics including animal weight (53 ±2kg), spleen weight, and preparation time (Table 1). Baseline lactate was similar amongst groups (p = 0.325). There was no significant difference in pH, hemodynamic parameters, blood counts, chemistries or viscoelastic parameters of coagulation (Table 1).

### Injury characteristics

Percent transection of the left lobe of the liver was not significant amongst groups (p = 0.980) and injury duration did not differ (p = 0.270). Intervention duration was significantly different between FP (12±2s) and BP (16±2s), p<0.001 (Table 2).

Hemodynamic parameters and lactate following injury did not differ among groups. Shed blood was not statistically different among groups (p = 0.454), Table 2.

**Table 1. Baseline characteristics for Free Pellets (FP), Bagged Pellets (BP) and no intervention (CON).**

| Variable | FP (n = 8) | BP (n = 8) | CON (n = 8) | p-value |
|---|---|---|---|---|
| Weight (kg) | 53 ± 2 | 53 ± 2 | 53 ± 2 | 0.935 |
| Total Prep (min) | 98 ± 23 | 93 ± 22 | 110 ± 29 | 0.387 |
| Normal Saline (mL) | 161 ± 48 | 148 ± 41 | 175 ± 56 | 0.555 |
| Spleen Weight (g) | 476.6 ± 120.6 | 502.4 ± 95.7 | 498.4 ± 103.5 | 0.875 |
| *Baseline Physiology* | | | | |
| Heart Rate (bpm) | 102 ± 16 | 96 ± 12 | 106 ± 26 | 0.563 |
| Systolic BP (mmHg) | 99 ± 9 | 94 ± 8 | 94 ± 9 | 0.329 |
| MAP (mmHg) | 66 ± 3 | 64 ± 6 | 67 ± 6 | 0.376 |
| Diastolic BP (mmHg) | 49 ± 4 | 50 ± 5 | 53 ± 5 | 0.373 |
| Central Venous Pressure (mmHg) | 6 ± 2 | 6 ± 2 | 7 ± 1 | 0.32 |
| EtCO$_2$ | 41 ± 1 | 41 ± 1 | 41 ± 1 | 0.943 |
| Systemic Vascular Resistance | 830 ± 77 | 869 ± 146 | 948 ± 172 | 0.238 |
| Cardiac Output (L/min) | 5.7 ± 0.6 | 5.5 ± 1.1 | 5.2 ± 1.1 | 0.548 |
| Mixed Venous Saturation | 62 ± 12 | 64 ± 13 | 67 ± 8 | 0.711 |
| Temperature (˚C) | 37.8 ± 0.8 | 37.7 ± 0.5 | 37.7 ± 0.7 | 0.859 |
| pO$_2$ (mmHg) | 78 ± 20 | 77 ± 15 | 79 ± 15 | 0.976 |
| pCO$_2$ (mmHg) | 42.0 ± 3.5 | 45.3 ± 3.9 | 42.3 ± 1.8 | 0.096 |
| Potassium (mmol/L) | 4.1 ± 0.3 | 4.1 ± 0.1 | 4.1 ± 0.4 | 0.987 |
| Ionized Calcium (mmol/L) | 1.29 ± 0.07 | 1.35 ± 0.04 | 1.29 ± 0.06 | 0.087 |
| pH | 7.454 ± 0.036 | 7.437 ± 0.041 | 7.435 ± 0.020 | 0.489 |
| Serum Lactate (mmol/L) | 2.0 ± 0.2 | 2.0 ± 0.7 | 1.7 ± 0.3 | 0.325 |
| Hgb (g/dL) | 10.3 ± 0.7 | 10.5 ± 0.9 | 10.3 ± 0.9 | 0.845 |
| Leukocytes | 20.71 ± 6.32 | 21.23 ± 5.46 | 20.97 ± 4.64 | 0.982 |
| Platelets | 319 ± 83 | 356 ± 113 | 331 ± 56 | 0.699 |
| Clotting Time (sec) | 718 ± 223 | 714 ± 238 | 853 ± 206 | 0.381 |
| Clot Formation Time (sec) | 241 ± 206 | 252 ± 121 | 333 ± 190 | 0.532 |
| α (˚) | 56 ± 16 | 53 ± 14 | 46 ± 17 | 0.447 |
| Maximum Clot Firmness (mm) | 60 ± 5 | 61 ± 7 | 59 ± 7 | 0.789 |
| LI30 (%) | 97 ± 2 | 99 ± 1 | 99 ± 1 | 0.078 |

All values are expressed as mean ± standard deviation.

### End pre-hospital (T = 180)

There was no significant difference in all physiologic measures including lactate, mean arterial pressure (MAP), end tidal CO$_2$, potassium, and hemoglobin levels. Intra-abdominal pressures did not differ between groups. Coagulation parameters did not differ among groups. All FP animals, 6/8 BP animals and 2/8 CON animals survived to simulated hospital arrival. In comparison to CON, survival was significantly higher for FP (p = 0.015) at 180 minutes. Over half of the CON animals expired prior to 2 hours, Fig 1.

### Damage control surgery

There was no difference observed in time to manual control of the liver hemorrhage among groups, however, FP manual control (215 ± 89s) was observed to be elevated in comparison to BP (165 ± 43s) and CON (65 ± 1s), however did not achieve statistical significance, p = 0.051 (Table 2). Damage control surgery duration was significantly longer in FP animals (18 ± 3min) in comparison to BP animals (9 ± 2min), p = 0.001(Table 2). Amount of Fresh Whole Blood (FWB) transfused during DCS did not differ, p = 0.958 (Table 2).

**Table 2. Injury, damage control surgery, and resuscitation characteristics for Free Pellets (FP), Bagged Pellets (BP) and no intervention (CON).**

| Variable | FP (n = 8) | BP (n = 8) | CON (n = 8) | p-value |
|---|---|---|---|---|
| *Injury/Intervention Characteristics* | | | | |
| Injury Duration (sec) | 39 ± 8 | 38 ± 8 | 45 ± 11 | 0.270 |
| Transection (%) | 69.5 ± 8.4 | 69.2 ± 9.4 | 68.7 ± 8.2 | 0.980 |
| Shed Blood (mL) | 1757.2 ± 381.3 | 1838.0 ± 360.8 | 2014.9 ± 482.7 | 0.454 |
| Intervention Duration (sec) | 12 ± 2 | 16 ± 2 | - | <0.001 |
| Systolic BP (mmHg) | 52 ± 21 | 36 ± 14 | 50 ± 17 | 0.150 |
| MAP (mmHg) | 32 ± 12 | 22 ± 10 | 32 ± 10 | 0.109 |
| EtCO$_2$ | 29 ± 9 | 21 ± 8 | 28 ± 7 | 0.139 |
| Lactate (mmol/L) | 3.0 ± 1.1 | 3.5 ± 0.8 | 3.0 ± 1.1 | 0.548 |
| *End Pre-Hospital (T = 180)* | | | | |
| All animals Hextend (mL) | 975 ± 212 | 1013 ± 223 | 900 ± 278 | 0.639 |
| Survivors Hextend (mL) | 975 ± 212 | 1100 ± 155 | 1050 ± 212 | 0.498 |
| *Damage Control Surgery (DCS)* | | | | |
| Manual Control (sec) | 215 ± 89 | 165 ± 43 | 65 ± 1 | 0.051 |
| DCS Duration (min) | 18 ± 3* | 9 ± 2 | 14 ± 9 | 0.001 |
| Missed Pellets (animals) | 2 | 0 | - | |
| Average Missed Pellets (n) | 6.5 | - | - | |
| Fresh Whole Blood (mL) | 831 ± 175 | 806 ± 135 | 823 ± 173 | 0.958 |
| *Intensive Care Phase* | | | | |
| Fresh Frozen Plasma (mL) | 563 ± 259 | 417 ± 303 | 500 | 0.35 |
| 50% Glucose (mL) | 88 ± 44 | 100 ± 32 | 150 | 0.156 |
| Insulin/50% Glucose (n) | 2 ± 1 | 2 ± 1 | 3 | 0.049 |
| 8.4% Sodium Bicarbonate (n) | 0* | 0* | 2 | <0.001 |
| Epinephrine (mg) | 0.25 ± 0.35* | 0.17 ± 0.30* | 3.25 | <0.001 |
| Norepinephrine (mg) | 0 | 383 ± 938 | 105.6 | 0.263 |
| Lactated Ringers (mL) | 681 ± 47 | 670 ± 134 | 687 | 0.816 |

All values are expressed as mean ± standard deviation.

*demonstrates significance in comparison to CON with p <0.05.

## Intensive care phase

There was no significant difference detected in the amount of LR or FFP administered among groups (Table 2). Only one CON animal survived to the end of the study while all FP and BP

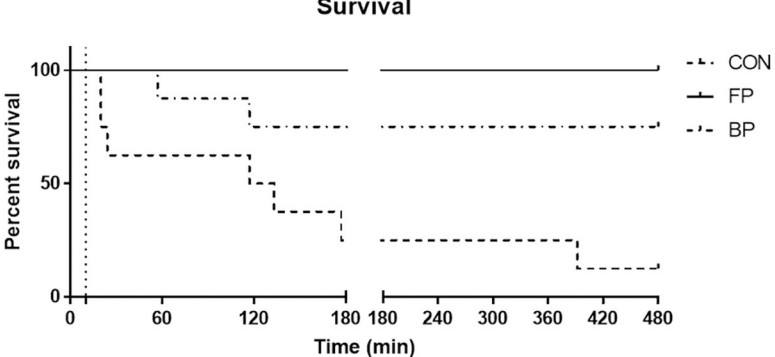

**Fig 1. Kaplan-Meier survival curve at 480 minutes.**

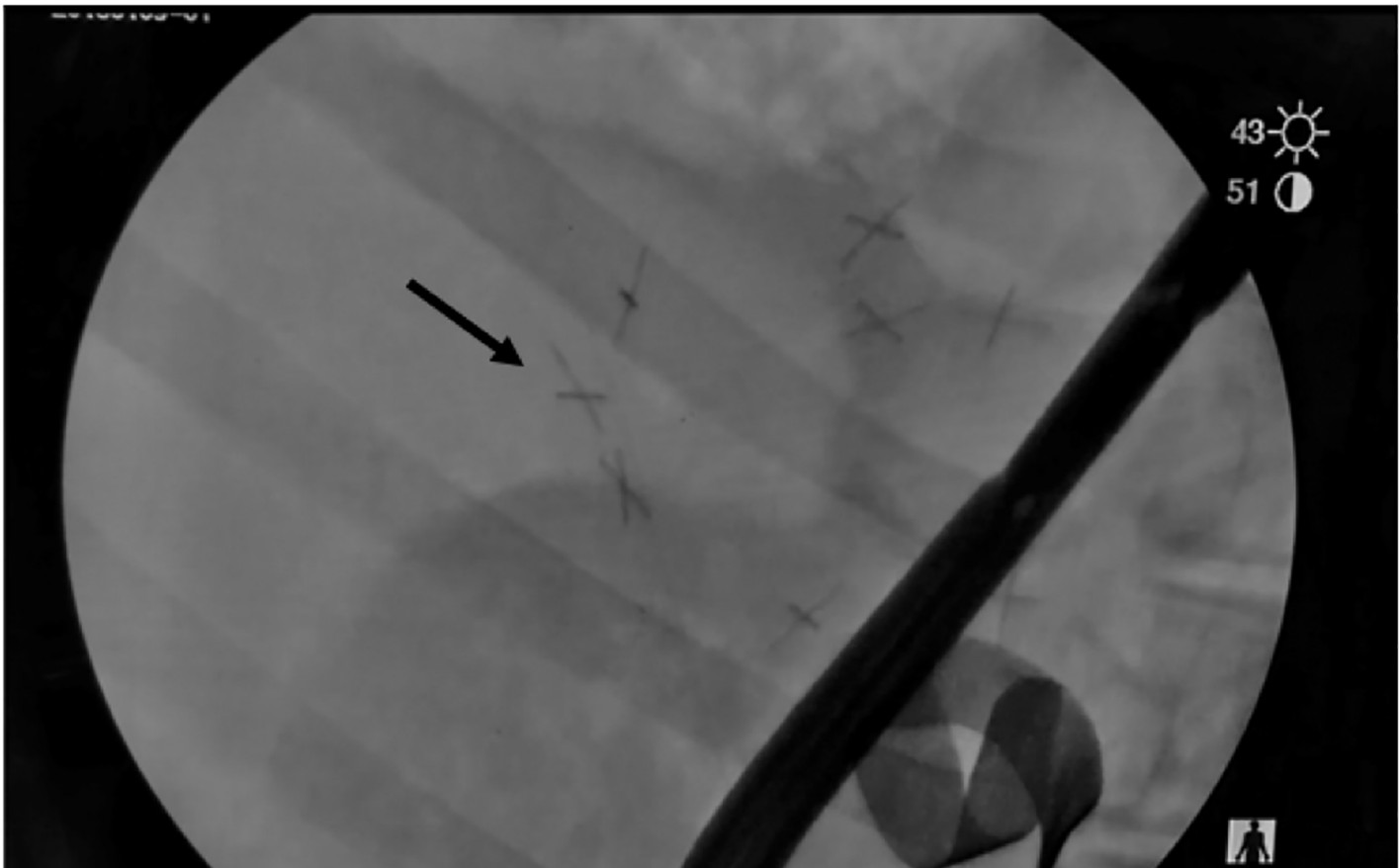

**Fig 2. X-ray image of retained pellets within the abdomen following damage control surgery.** Arrow denotes one of seven pellets with radiopaque X.

animals that survived simulated pre-hospital, survived to the end of the study. The surviving CON animal required more CON and sodium bicarbonate in comparison to the FP and BP groups during the ICU period (Table 2). There was no significant difference in pressor requirement between FP and BP groups, nor was there a difference in glucose and Insulin/ D50. Upon fluoroscopy of the abdomen, there were two animals with documented pellets in the abdomen that were not retrieved at the time of DCS via visual inspection (7 pellets in one animal and 4 in the other animal), Fig 2. All pellets identified by radiologic imaging were successfully identified and removed during necropsy. There were no remaining bags noted during fluoroscopy in the BP group.

## End study (T = 480)

Both FP and BP had significantly improved survival in comparison to CON at 480 minutes (p = 0.003 and p = 0.018, respectively), Fig 1. With exception of survival, statistical comparisons of all three groups was not possible due to CON sample size (n = 1) and therefore comparisons were made only between FP and BP groups. Lactate levels trended towards baseline values in both the FP and BP groups (Fig 3). There was no significant difference in ischemic pathology in the terminal ileum, kidney, pancreas, liver, lung or cardiac apex amongst all groups. However of note, we observed ischemic changes in the terminal ileum as evidence by apoptotic bodies in the presence or absence of mucosal ulcerations (FP 8/8; BP 7/8). For the

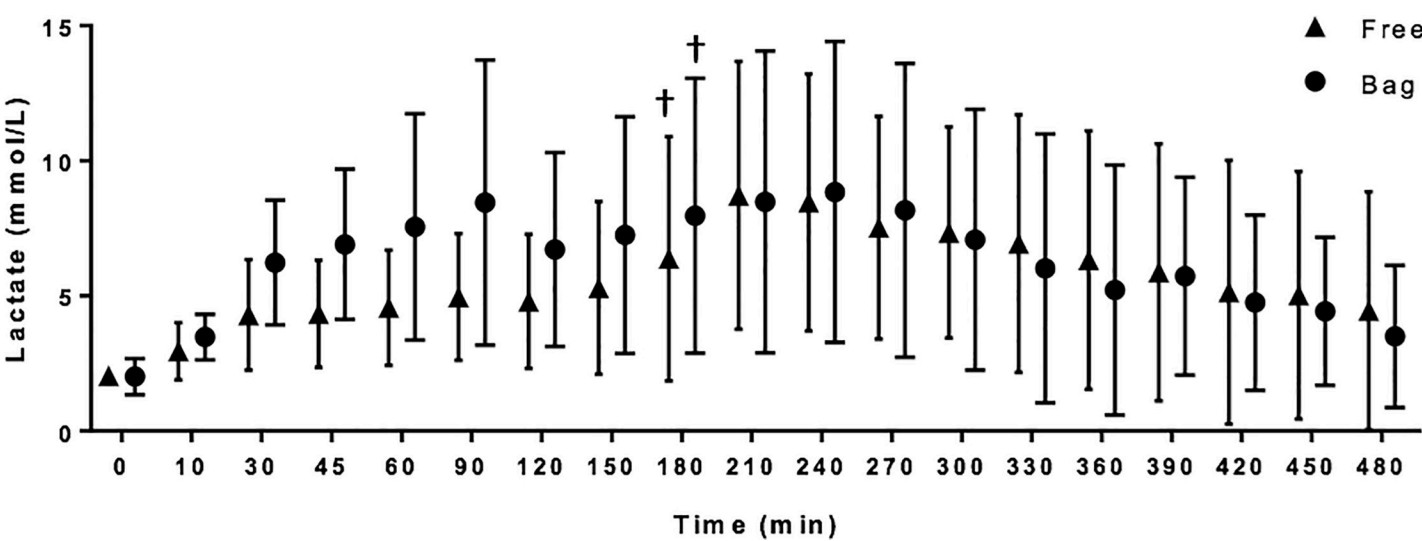

**Fig 3. Serum lactate.** † denotes significance from respective baseline value.

two animals with pellets in the abdomen for a full eight hours, there was no evidence of adhesions or serosal injury to the bowel.

## Discussion

This study is the first to demonstrate a novel off-label use for the use of XSTAT 30™ in mitigation of non-compressible abdominal hemorrhage in the setting of simulated prolonged pre-hospital care. Not only did we observe improved survival with deployment of XSTAT 30™ in both the bagged and free pellet formats in comparison to CON, but additionally the physiologic penalty incurred during injury and simulated pre-hospital period was reversible with modern damage control techniques. While a great deal more translational evidence will be required to bring XSTAT 30™ to the clinical armamentarium it does not diminish the fact that the availability of any non-endovascular-based intervention for pre-hospital NCTH management has the potential to revolutionize combat casualty care in Anti-access Area Denial theaters of operation, particularly those where immediate tactical evacuation to Role II or Role III care is not available. The potential for application in austere civilian environments is significant including the stabilization of patients with NCTH arriving to Level IV & V trauma centers that subsequently required transport to higher echelons of care.

One of most interesting features of this study was the indeterminate mechanism of action for XSTAT 30™. We predicted that intracavitary pressures at or near the site of injury would be elevated as previously observed with alternative intracavitary or external adjuncts for NCTH management [12]. Despite an observable space-filling effect of expanded sponges at the wound site, concomitant increases of pressure were not observed. Additionally, confounding was the lack in any decrease in intracavitary shed blood volumes in animals that received XSTAT 30™ in either form as compared to CON alone. We therefore surmised that the mechanism for XSTAT 30™ in the management of NCTH must be highly localized.

It is reasonable to predict that XSTAT 30™ control of hemorrhage at the site of injury is discrete and that it may enhance and localize concentration of blood products due to the rapid absorption of serum during sponge expansion, promoting more effective and robust clot formation. With either mechanistic theory, XSTAT 30™ would likely minimize reperfusion injury incurred with use of current treatments for NCTH, as any ischemia to tissues impacted by

XSTAT 30™ use will also be discrete and localized [6, 13]. This imparts a significant benefit over the use of aortic occlusion systems like Resuscitative Endovascular Balloon Occlusion of the Aorta (REBOA) for the management of NCTH in extended pre-hospital periods as the secondary injury and physiologic derangements that is observed with prolonged occlusion is mitigated [5, 6].

The fundamental limitation of endovascular approaches to pre-hospital NCTH management revolves around the narrow window of therapeutic aortic occlusion time before critical metabolic thresholds for secondary injury are exceeded [14]. We must additionally temper these results in the context that the study was large-animal translational in design and blinding of the investigators to the intervention was not possible. However, this study exemplifies the need for a departure from complete endovascular occlusion for extended pre-hospital transport times in casualties where control of non-compressible bleeding is the principal concern.

Here we used a model of simulated pre-hospital care for 180 minutes. This is a substantial increase in pre-hospital duration from our previous study using REBOA in the same model [15]. Despite the 2 hour increase in pre-hospital care in the absence of balanced blood product resuscitation, the metabolic burden of the animals in the current study at initiation of DCS was half that seen in the animals that underwent REBOA for one third of the total pre-hospital time in our previous work [15]. Modified applications of the REBOA catheter have been developed to extend its usefulness in prolonged field care, specifically intermittent REBOA. Kuckelman et al. have demonstrated usefulness of intermitment REBOA in a similar solid organ injury model with 100% survival though only up to two-hours of simulated pre-hospital observation [16]. While investigators continue to refine endovascular techniques for NCTH the authors want to emphasize that perhaps a non-endovascular approach may be better suited to austere environments.

While endovascular occlusion of the aorta is the prominent technological advance for abdominal NCTH in recent years, alternatives to this approach have also seen significant investment including expanding intracavitary foams [12]. These products address NCTH through space filling and/or acute increases in intracavitary pressure [12]. While this approach has merit in the context of extended field care there are significant complications to include serosal injury to the bowel and adhesive disease with prolonged exposure [12]. Interestingly, in the execution of this study some XSTAT 30™ pellets remained in the abdomen following DCS and initial repair of the injury despite a thorough examination of the abdomen, however at the end of the 8 hour study we did not identify any serosal injury or adhesions. In addition, the risk of retained pellets is low given the radiopaque markers on both free and bagged pellets, which were easily identified by radiographic evidence.

This study generated evidence that free XSTAT 30™ pellets have the potential to increase time for access to the injury site for definitive control. XSTAT 30™ devices with pellet-containing radiopaque bags were created to circumvent the need for individual pellet retrieval and streamline injury site access. While this design change to XSTAT 30™ pellet application addresses potential issues regarding pellet retrieval time, in this study it led to a reduction in overall survival likely attributed to changes in pellet liquid absorption and expansion dynamics though we did not collect or measure indices that would prove or disprove this theory. However, as demonstrated in a postpartum hemorrhage ewe model which used pellets within a bag, there was no mention of saturation or expansion dysfunction [9]. Rodriguez et al also demonstrated 100% survival in all ewes that received this intervention of bagged pellets during postpartum hemorrhage, which differs from our current results [9].

This study is not without limitation. The placement of the bespoke XSTAT 30™ trocar, which was located in the epigastric region medial to where the liver laceration would be created is not necessarily comparable to a penetrating traumatic injury that may have a different

trajectory. Future studies will need to be implemented to observe the migration of XSTAT 30™ pellets to the bleeding source if placed in a different quadrant within the abdomen. Secondly, there is no current method of placement for prehospital application. The proposed mechanism of deployment would be similar to the administration of intraabdominal foam or diagnostic peritoneal lavage, in which the provider would perform abdominal cutdown for trocar placement, however, this still needs to be tested. Third, our model employed Hextend as a control, which is no longer the gold standard for pre-hospital resuscitation used in our previous studies of NCTH interventions and may not be appropriate for future studies in the context for prolonged field care applications. Despite the "black box" warning associated with Hextend, the investigators selected Hextend as a pre-hospital positive control to maintain continuity in our current program of research which includes multiple and varying adjuncts for NCTH. In our opinion this selection was also appropriate in order to coincide with the current Tactical Combat Casualty Care guidelines when blood products are not available (worst-case scenario for pre-hospital resuscitation) [1]. While these guidelines incorporate FWB or 1:1:1 resuscitation as the ideal fluid of choice, it may not be available in an austere environment or rural hospital [17].

## Conclusion

XSTAT 30™ may be a viable intervention to address NCTH in prolonged pre-hospital care without secondary ischemic complications and metabolic derangements as demonstrated by improved survival in comparison to fluid resuscitation alone. XSTAT 30™ pellets can be identified, either visually or upon radiographic imaging, and removed prior to definitive abdominal closure in the laboratory setting. Furthermore, continued development of XSTAT 30™ within bags is a potential avenue for improvement in pellet retrieval and reduction in total operative time for damage control surgery. Future studies will aim to determine any infectious or harmful effects with prolonged exposure of pellets within the abdomen and will also include further development of XSTAT 30™ within bags along with an intraperitoneal introducer device. Despite these promising results, there is a great deal of continued research and development required to move XSTAT 30™ from the experimental to operational application as a solution set for NCTH and off label use should not be considered until further research is complete.

## Supporting information

**S1 Fig. Depiction of XSTAT 30™ with Free Pellets (FP) and pellets in radiopaque bags (BP).** Subpanel a: Compact and expanded pellets in radiopaque bags. Subpanel b: Pellets in radiopaque bags in-situ. Subpanel c: Compact free pellets. Subpanel d: Expanded free pellets. Subpanel e: Expanded free pellets in-situ.
(TIFF)

## Acknowledgments

We would like to acknowledge the work of our veterinary pathologist Rosemary Makar for her contribution to this study.

## Author Contributions

**Conceptualization:** James D. Ross.

**Data curation:** Todd L. Graham, Lauren N. Wilson.

**Formal analysis:** Todd L. Graham, James D. Ross.

**Funding acquisition:** James D. Ross.

**Investigation:** Alicia M. Bonanno, Todd L. Graham, Lauren N. Wilson, James D. Ross.

**Methodology:** Alicia M. Bonanno, Todd L. Graham, Lauren N. Wilson, James D. Ross.

**Project administration:** Todd L. Graham, James D. Ross.

**Resources:** James D. Ross.

**Software:** Todd L. Graham, James D. Ross.

**Supervision:** Todd L. Graham, James D. Ross.

**Validation:** James D. Ross.

**Visualization:** James D. Ross.

**Writing – original draft:** Alicia M. Bonanno.

**Writing – review & editing:** Alicia M. Bonanno, Todd L. Graham, Lauren N. Wilson, James D. Ross.

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
