## [Decision Letter · Decision Letter 0]

12 Sep 2019

PONE-D-19-14707

Novel Use of XSTAT 30™ for Mitigation of Lethal Non-Compressible Torso Hemorrhage in Swine

PLOS ONE

Dear Dr Ross,

Thank you for submitting your manuscript to PLOS ONE. After careful consideration, we feel that it has merit but does not fully meet PLOS ONE’s publication criteria as it currently stands. Therefore, we invite you to submit a revised version of the manuscript that addresses the points raised during the review process.

We would appreciate receiving your revised manuscript by Oct 27 2019 11:59PM. To enhance the reproducibility of your results, we recommend that if applicable you deposit your laboratory protocols in protocols.io, where a protocol can be assigned its own identifier (DOI) such that it can be cited independently in the future. For instructions see: http://journals.plos.org/plosone/s/submission-guidelines#loc-laboratory-protocols

We look forward to receiving your revised manuscript.

Kind regards,

Zsolt J. Balogh, MD, PhD, FRACS

Academic Editor

PLOS ONE

Journal Requirements:

2. At this time, we request that you  please report additional details in your Methods section regarding animal care, as per our editorial guidelines: 1) Please provide the source of the animals, 2) Please explain whether the animals gained consciousness at any point during the experiment or if they were anaesthetised through out the experiment until the time of euthanasia or death. Thank you for your attention to these requests.

Additional Editor Comments (if provided):

Reviewers' comments:

Reviewer's Responses to Questions

**Comments to the Author**

1. Is the manuscript technically sound, and do the data support the conclusions?

Reviewer #1: No

Reviewer #2: Yes

2. Has the statistical analysis been performed appropriately and rigorously? 

Reviewer #1: Yes

Reviewer #2: I Don't Know

3. Have the authors made all data underlying the findings in their manuscript fully available?

Reviewer #1: Yes

Reviewer #2: Yes

4. Is the manuscript presented in an intelligible fashion and written in standard English?

Reviewer #1: Yes

Reviewer #2: Yes

5. Review Comments to the Author

Reviewer #1: The authors have described an animal experiment where Xstat was used for NCTH.

I have several comments

1. Why was Hextend used as a control group? Its use is not recommended by TCCC, whole blood would have been the correct comparison group.

2. How often does a penetrating wound have a 30 mm opening? This is an important part of the model as access to the peritoneal cavity will likely be a limiting feature in the prolonged field care environment.

3. Are the number of pellets the same between the FP and BP groups?

4. The 30 mm part was placed in the upper abdomen, right over the liver injury, facilitating the pressure action of the pellets. As described by the authos this is a serious limitation of the model. In the prehospital setting the medics will not know if the bleeding site is right below a skin wound, in fact there is little relation of the superficial wound to the actual bleeding organ/vessel.

5. The authors state “Future studies will need to be implemented to observe the migration of XSTAT 30™ pellets to the bleeding source if placed in a different quadrant within the abdomen.” Why do the authors feel the pellets will migrate, unless there is a new capability of these pellets, please remove this statement.

6. Before any suggestion of use of these pellets for NCTH, the authors must perform studies were the pellets are placed remotely from the injury.

7. The authors have studies pellet function in low pressure liver injury model, the easiest to control. Before any recommendation of use, the authors must study the use of pellets in a high pressure iliac artery (lower quadrant) injury model with pellets placed remotely (upper quadrant). In other words create the worst case scenario versus the best.

8. The first sentence of the conclusion and that of the abstract is irresponsible. “XSTAT 30™ may be a viable intervention to address NCTH in prolonged pre-hospital care without secondary ischemic complications and metabolic dyshomeostasis as demonstrated by improved survival in comparison to fluid resuscitation alone.” As stated above the authors have created a model of the best case, pellets placed right over the liver with a “bespoke”30 mm trocar. Before any suggestion that this is ready for battlefield use this must be replicated in studies that more accurately reflect prehospital conditions.

Reviewer #2: Thank you for an excellent piece of work, very relevant and very well described.

I have two questions which I think need further detail in your write up (first two) and two which are non critical, just need clarification although you may have already described these elsewhere (last two).

1. how exactly are the pellets inserted? you describe towards the end that they went in through the port close to the liver injury and that insertion elsewhere in other quadrants is yet to be tested, but a little more detail as to how exactly the operator inserts them (and maybe some pictures) would help the reader understand how this might be feasible in the future with a PHC applicator. Also what do they look like in and out of the bags? Please ignore this if described elsewhere, but maybe reference the description?

2. Regarding the comment “There was no significant difference in ischemic pathology in the terminal ileum, kidney, pancreas, liver, lung or cardiac apex amongst all groups.”.... does this mean there was no damage in all groups or that there was some damage but no difference? If the latter, it would be useful to see a description of any ischaemic changes noted.

3. do retained pellets cause any harm or are they reabsorbed? (you may have already described this in a previous paper).

4. no difference in shed blood: did this include the pigs that died? did they have laparotomy and measurement of shed blood, i would presume so but didn’t see it described.

Thanks

6. PLOS authors have the option to publish the peer review history of their article (what does this mean?). If published, this will include your full peer review and any attached files.

Reviewer #1: No

Reviewer #2: Yes: Dr Samy Sadek

---

## [Author Response · Author response to Decision Letter 0]

8 Jan 2020

Reviewer 1 - Comment 1: Why was Hextend used as a control group? Its use is not recommended by TCCC, whole blood would have been the correct comparison group.

Authors’ Response 1: Hextend was used as a control group for two reasons. The first is that, Hextend represents the “worst case scenario” for resuscitation fluid in the field. If a device is effective in managing NCTH in the presence of a clear fluid resuscitation then it is reasonable to hypothesize that it would be effective in combination with resuscitation fluids like whole blood, plasma, or a multi-function resuscitation fluid. The second reason is that Hextend was used as the control for research conducted in the efficacy of different devices for the management of NCTH including REBOA and the Abdominal Aortic Junctional Tourniquet Torso Plate. The continued used of Hextend in these types of studies allows for comparison with historical studies of device efficacy. Hextend is still a component of the TCCC guidelines for resuscitation in the absence of availability of whole blood. This was addressed in the discussion of the original manuscript discussion. It is also of note that the United States Army Special Operations Command Combat Development Directorate was aware of and concurred with the design of this study.

Comment 2: How often does a penetrating wound have a 30 mm opening? This is an important part of the model as access to the peritoneal cavity will likely be a limiting feature in the prolonged field care environment.

Authors’ Response 2: Entry and exit would sizes are variable dependent on the type of ordinance or fragment that causes the injury. The authors recognize that access to the peritoneal cavity may be variable dependent on injury, however, RevMedx is designing an access device to accompany the XSTAT30 for use in NCTH. XSTAT30 is another tool in the armamentarium for treatment of NCTH pre-hospital and may not necessarily be appropriate for all comers.

Comment 3: Are the number of pellets the same between the FP and BP groups?

Authors’ Response 3: Yes.

Comment 4: The 30 mm part was placed in the upper abdomen, right over the liver injury, facilitating the pressure action of the pellets. As described by the authos this is a serious limitation of the model. In the prehospital setting the medics will not know if the bleeding site is right below a skin wound, in fact there is little relation of the superficial wound to the actual bleeding organ/vessel.

Authors’ Response 4: The authors concur that there is little relation of superficial would to potential intra-peritoneal bleeding or organ injury. The port was not placed directly above the liver injury. The port was placed midline while the injury was left lateral and deep to the port site. In addition, the port could not have facilitated pressure action of the pellets as the pellets in the tested application do not function via application of pressure as determined by placement of pressure transducers near the site of the injury. The authors suspect that the sponges do not require immediate delivery in direct proximity to the injury site as evidenced by their migration toward the site of bleeding from the application site.

Comment 5: The authors state “Future studies will need to be implemented to observe the migration of XSTAT 30™ pellets to the bleeding source if placed in a different quadrant within the abdomen.” Why do the authors feel the pellets will migrate, unless there is a new capability of these pellets, please remove this statement.

Authors’ Response 5: As changes in intra-abdominal pressure were not observed at the injury site in response to sponge expansion, the authors hypothesize that a different mechanism of action is at play. Specifically, the authors observed sponges in contact with the injury site despite the fact that they were not deployed directly above the site of injury. We suspect that as the pellets absorb plasma and expand that they move countercurrent to the source of bleeding. The concentration of red cells and platelets at the site of bleeding through absorption of plasma by the sponges may speed clotting and provide a matrix from which clot could propagate and would be reinforced (analogous to rebar in concrete). The authors intend to study this phenomenon in vivo.

Comment 6: Before any suggestion of use of these pellets for NCTH, the authors must perform studies were the pellets are placed remotely from the injury.

Authors’ Response 6: The authors concur. An assessment of efficacy of the XSTAT sponges when placed further from the injury site would be interesting, particularly in the context of mechanism of action. 

Comment 7: The authors have studies pellet function in low pressure liver injury model, the easiest to control. Before any recommendation of use, the authors must study the use of pellets in a high pressure iliac artery (lower quadrant) injury model with pellets placed remotely (upper quadrant). In other words create the worst case scenario versus the best.

Authors’ Response 7: The authors concur that an exploration of XSTAT use in a named arterial model of NCTH would be beneficial, both to understanding its range of efficacy and to better understanding its mechanism of action and the potential for sponge migration to the source of bleeding. We would however comment that the liver injury model employed in this experiment is both venous and arterial in nature. The left lateral lobe transection model described results in the transection of two significant arterioles that have been measured at up to 3mm in diameter and result in significant arterial bleeding. We 

Comment 8: The first sentence of the conclusion and that of the abstract is irresponsible. “XSTAT 30™ may be a viable intervention to address NCTH in prolonged pre-hospital care without secondary ischemic complications and metabolic dyshomeostasis as demonstrated by improved survival in comparison to fluid resuscitation alone.” As stated above the authors have created a model of the best case, pellets placed right over the liver with a “bespoke”30 mm trocar. Before any suggestion that this is ready for battlefield use this must be replicated in studies that more accurately reflect prehospital conditions.

Authors’ Response 8: The authors opine that the statement is accurate as written – the operating word being “may”. The principal investigator for the authors’ laboratory has investigated numerous interventions for NCTH and hemorrhage-induced cardiac arrest including REBOA, SAAP, AAJT and XSTAT and in the context of those investigations is confident in the interpretation of the results of this study and the opinions that XSTAT may be a viable intervention for NCTH. However, the authors do not want readers to overestimate the device’s readiness for battlefield application, nor would we condone or suggest its use off-label by deployed medics therefore we have expanded the paragraph appropriately to include caution. 

Additionally, the authors would argue that this model reflects the “best case”. The experiment was designed to limit confounding variables in application method and timing. While the trocar was bespoke, it was not placed directly over the site of the injury. There was in fact a measurable distance between the midline and site of injury. In addition, the injury was not in plane with the direction of application. The transection portion of the liver and site of bleeding was transverse to the application of sponges from the XSTAT. In addition, the transection portion of the left lobe remained in anatomical alignment with the rest of the liver, limiting direct access of sponges to the site of bleeding. The authors concur that further study is required and as such, studies are ongoing.

Reviewer 2 - Comment 1: how exactly are the pellets inserted? you describe towards the end that they went in through the port close to the liver injury and that insertion elsewhere in other quadrants is yet to be tested, but a little more detail as to how exactly the operator inserts them (and maybe some pictures) would help the reader understand how this might be feasible in the future with a PHC applicator. Also what do they look like in and out of the bags? Please ignore this if described elsewhere, but maybe reference the description?

Authors’ Response 1: XSTAT sponges were deployed per manufacturer’s instruction through a 30mm trocar placed at the midline and lateral to the site of liver injury. The methods have been revised to better clarify mode of insertion and deployment. We have added a supplemental figure that shows the deployed pellets in free and bagged forms in situ.

Comment 2: Regarding the comment “There was no significant difference in ischemic pathology in the terminal ileum, kidney, pancreas, liver, lung or cardiac apex amongst all groups.”.... does this mean there was no damage in all groups or that there was some damage but no difference? If the latter, it would be useful to see a description of any ischaemic changes noted.

Authors’ Response 2: As only 1 Hextend animal survived to end of study we were unable to make comparisons to the treated groups. In the case of the treated groups, ischemia injury was of note in the terminal ileum and has been better described in the revisions. The authors are unclear as to the source of the ischemia as SHAM controls were not included in the experimental design.

Comment 3: do retained pellets cause any harm or are they reabsorbed? (you may have already described this in a previous paper).

Authors’ Response 3: At this time there is a lack of data to answer the reviewer’s question. The sponges have been produced in a bioabsorbable form but have not been tested. Investigation into the impact of pellet retention is ongoing.

---

## [Decision Letter · Decision Letter 1]

9 Apr 2020

PONE-D-19-14707R1

Novel Use of XSTAT 30™ for Mitigation of Lethal Non-Compressible Torso Hemorrhage in Swine

PLOS ONE

Dear Dr Ross,

Thank you for submitting your manuscript to PLOS ONE. After careful consideration, we feel that it has merit but does not fully meet PLOS ONE’s publication criteria as it currently stands. Therefore, we invite you to submit a revised version of the manuscript that addresses the points raised during the review process.

We would appreciate receiving your revised manuscript by May 24 2020 11:59PM. To enhance the reproducibility of your results, we recommend that if applicable you deposit your laboratory protocols in protocols.io, where a protocol can be assigned its own identifier (DOI) such that it can be cited independently in the future. For instructions see: http://journals.plos.org/plosone/s/submission-guidelines#loc-laboratory-protocols

We look forward to receiving your revised manuscript.

Kind regards,

Zsolt J. Balogh, MD, PhD, FRACS

Academic Editor

PLOS ONE

Reviewers' comments:

Reviewer's Responses to Questions

**Comments to the Author**

1. If the authors have adequately addressed your comments raised in a previous round of review and you feel that this manuscript is now acceptable for publication, you may indicate that here to bypass the “Comments to the Author” section, enter your conflict of interest statement in the “Confidential to Editor” section, and submit your "Accept" recommendation.

Reviewer #1: (No Response)

2. Is the manuscript technically sound, and do the data support the conclusions?

Reviewer #1: No

3. Has the statistical analysis been performed appropriately and rigorously? 

Reviewer #1: Yes

4. Have the authors made all data underlying the findings in their manuscript fully available?

Reviewer #1: Yes

5. Is the manuscript presented in an intelligible fashion and written in standard English?

Reviewer #1: Yes

6. Review Comments to the Author

Reviewer #1: The authors have responded. However they argue that this intervention is revolutionary. see the sentence in the discussion. " The availability of a non-endovascular-based intervention for pre-hospital NCTH management has the potential to revolutionize combat casualty care in Anti-access Area Denial theaters of operation, particularly those where immediate tactical evacuation to Role II or Role III care is not available." This reviewer is extremely familiar with swine anatomy and i have personally cared for thousands of casualties in the prehospital and hospital combat casualty care area. This approach has promise. However the authors must temper their assessment of its potential usefulness, until further work is done.

They need to articulate in the abstract and the discussion the many and significant limitations of this approach. The experiment is sound and I do not have any issues with the data driven conclusions. There is promise here. However I do have significant issue with the tenor of this paper as currently written. The OHSU trauma group, led by Dr Schreiber has significant experience in hemorrhage control studies. I would suggest including in the group a physician that has actually taken care of trauma patients, at the faculty level, and preferably who has deployed to the combat setting. They have recently presented swine work with 3 hour survival with a competing approach, and have presented those data in a very balanced fashion. Until this approach is evaluated in an arterial injury and with the 30 mm bespoke trocar placed as far away as possible from the injury, this can not in any way be recommended. The problem with this is that the pellets are currently cleared for an extremely limited intervention. Medics could read this paper and deploy them tomorrow. The bag, according to these data is not necessary. I consider even the suggestion that this is an intervention to be considered is irresponsible. Additionally, placing a 30 mm trocar into the abdomen of a trauma patient is an advanced surgical procedure. It will require surgical level anesthesia and intubation.

The authors compare the value of their approach to REBOA, using very old references. (ref 15), if they are going to do this type of comparison, they need to include modern references that describe partial or intermittent occlusion. I would suggest they delete most of this discussion as they do not have comparative data.

The authors need to add to the limitations that they were not blinded to the intervention and thus the OR and ICU care could be biased.

Obviously I strongly agree with the last sentence of the paper and feel it should be replicated in the abstract... there is a great deal of continued research and development required to move XSTAT 30™ from the experimental to operational application as a solution set for NCTH and off label use should not be considered until further research is complete.

The senior author has great experience in the lab. this is a well done study with interesting results. Please just review the paper, temper the discussion and in no way encourage any use until more demanding studies are completed.

7. PLOS authors have the option to publish the peer review history of their article (what does this mean?). If published, this will include your full peer review and any attached files.

Reviewer #1: No

---

## [Author Response · Author response to Decision Letter 1]

16 Jun 2020

We have revised the manuscript along the overarching theme of concerns presented by the reviewer and therefore did not do a point by point response. We would also like to thank the reviewer for their continued thoughtful comments and questions.

---

## [Decision Letter · Decision Letter 2]

22 Jul 2020

PONE-D-19-14707R2

Novel Use of XSTAT 30™ for Mitigation of Lethal Non-Compressible Torso Hemorrhage in Swine

PLOS ONE

Dear Dr. Ross,

Thank you for submitting your manuscript to PLOS ONE. After careful consideration, we feel that it has merit but does not fully meet PLOS ONE’s publication criteria as it currently stands. Therefore, we invite you to submit a revised version of the manuscript that addresses the points raised during the review process.

We look forward to receiving your revised manuscript.

Kind regards,

Zsolt J. Balogh, MD, PhD, FRACS

Academic Editor

PLOS ONE

Reviewers' comments:

Reviewer's Responses to Questions

**Comments to the Author**

1. If the authors have adequately addressed your comments raised in a previous round of review and you feel that this manuscript is now acceptable for publication, you may indicate that here to bypass the “Comments to the Author” section, enter your conflict of interest statement in the “Confidential to Editor” section, and submit your "Accept" recommendation.

Reviewer #1: (No Response)

2. Is the manuscript technically sound, and do the data support the conclusions?

Reviewer #1: Partly

3. Has the statistical analysis been performed appropriately and rigorously? 

Reviewer #1: Yes

4. Have the authors made all data underlying the findings in their manuscript fully available?

Reviewer #1: Yes

5. Is the manuscript presented in an intelligible fashion and written in standard English?

Reviewer #1: Yes

6. Review Comments to the Author

Reviewer #1: the authors have chosen to not highlight their responses, which is unusual.

Since they choose to continue to compare their approach to REBOA and cite animal studies that were published in 2014, it is appropriate to suggest they read and cite more recent data which directly address the issues they raise, and safely extend the survival out to 240 minutes. One of them is from their own Division.

Zilberman-Rudenko J, Behrens B, McCully B, et al. Use of Bilobed Partial Resuscitative Endovascular Balloon Occlusion of the Aorta (pREBOA) is Logistically Superior in Prolonged Management of a Highly Lethal Aortic Injury. J Trauma Acute Care Surg. 2020;10.1097

Kuckelman JP, Barron M, Moe D, et al. Extending the golden hour for Zone 1 resuscitative endovascular balloon occlusion of the aorta: Improved survival and reperfusion injury with intermittent versus continuous resuscitative endovascular balloon occlusion of the aorta of the aorta in a porcine severe truncal hemorrhage model. J Trauma Acute Care Surg. 2018;85(2):318-326.

7. PLOS authors have the option to publish the peer review history of their article (what does this mean?). If published, this will include your full peer review and any attached files.

Reviewer #1: No

---

## [Author Response · Author response to Decision Letter 2]

27 Aug 2020

Reviewer Comment 1 - the authors have chosen to not highlight their responses, which is unusual.

Authors’ Response 1 – the authors submitted a set of revisions with “track changes” enabled.

Reviewer Comment 2 - Since they choose to continue to compare their approach to REBOA and cite animal studies that were published in 2014, it is appropriate to suggest they read and cite more recent data which directly address the issues they raise, and safely extend the survival out to 240 minutes. One of them is from their own Division.

Zilberman-Rudenko J, Behrens B, McCully B, et al. Use of Bilobed Partial Resuscitative Endovascular Balloon Occlusion of the Aorta (pREBOA) is Logistically Superior in Prolonged Management of a Highly Lethal Aortic Injury. J Trauma Acute Care Surg. 2020;10.1097

Kuckelman JP, Barron M, Moe D, et al. Extending the golden hour for Zone 1 resuscitative endovascular balloon occlusion of the aorta: Improved survival and reperfusion injury with intermittent versus continuous resuscitative endovascular balloon occlusion of the aorta of the aorta in a porcine severe truncal hemorrhage model. J Trauma Acute Care Surg. 2018;85(2):318-326.

Authors’ Response 2 – The reviewer is correct in that the authors compare intra-abdominal XSTAT use to REBOA. However, partial REBOA, EVAC and intermittent REBOA are in the early stages of development and are not practically applicable to the first responder or medic at this time and furthermore do not necessarily add context to the discussion of XSTAT as described in this study.

The authors are aware of the Zilberman-Rudenko publication. The authors reject the reference as the experimental design is seriously flawed. An aortic punch model of hemorrhage is irrelevant regarding REBOA as it is a well-established clinical contraindication for the use of REBOA. The authors have discussed this issue with various other clinical and research thought-leaders in trauma and are confident in this choice.

Authors’ Response 3 – We have added to the discussion to include mention of alternative implementations of REBOA including a more recent Kuckelman reference that uses a similar solid organ injury model.

---

## [Decision Letter · Decision Letter 3]

16 Oct 2020

PONE-D-19-14707R3

Novel Use of XSTAT 30™ for Mitigation of Lethal Non-Compressible Torso Hemorrhage in Swine

PLOS ONE

Dear Dr. Ross,

Thank you for submitting your manuscript to PLOS ONE. After careful consideration, we feel that it has merit but does not fully meet PLOS ONE’s publication criteria as it currently stands. Therefore, we invite you to submit a revised version of the manuscript that addresses the points raised during the review process.

The paper clearly has the potential to contribute to the field, and the reviewers have laid out a clear path to strengthen the manuscript and bring a perspective to the study and the broader field that could be quite important. I really encourage you to look more holistically at the work and current approaches and carefully consider the reviewers' suggestions. They are supportive, and you need to be responsive to the suggestions.

We look forward to receiving your revised manuscript.

Kind regards,

Erin Lavik

Academic Editor

PLOS ONE

Reviewers' comments:

Reviewer's Responses to Questions

**Comments to the Author**

1. If the authors have adequately addressed your comments raised in a previous round of review and you feel that this manuscript is now acceptable for publication, you may indicate that here to bypass the “Comments to the Author” section, enter your conflict of interest statement in the “Confidential to Editor” section, and submit your "Accept" recommendation.

Reviewer #1: (No Response)

Reviewer #3: (No Response)

2. Is the manuscript technically sound, and do the data support the conclusions?

Reviewer #1: No

Reviewer #3: Partly

3. Has the statistical analysis been performed appropriately and rigorously? 

Reviewer #1: Yes

Reviewer #3: Yes

4. Have the authors made all data underlying the findings in their manuscript fully available?

Reviewer #1: Yes

Reviewer #3: Yes

5. Is the manuscript presented in an intelligible fashion and written in standard English?

Reviewer #1: Yes

Reviewer #3: Yes

6. Review Comments to the Author

Reviewer #1: The authors have persisted in comparing their approach to reboa, yet have provided inadequate balance in their presentation. This is unfortunate, as comparing their experimental data to reboa is not needed in this precclinical paper. Lastly, when pacing a reboa, either in the ED or prehospital, it is unknown if an aortic injury is present.

Reviewer #3: Thank you for the opportunity to review this work by Bonanno and colleagues on a novel application of hemostatic pellets in an uncontrolled torso hemorrhage swine model. The authors have conducted a rigorous evaluation of this novel approach and have some very interesting results. I have a few comments and questions for the authors that I hope they will address.

1) Although I think I understand the meaning of the term "dyshomeostasis" I am not certain this is a very common term and would suggest that it is not specific enough to be considered an endpoint for your 2nd hypothesis. I would suggest using a different term to convey your meaning.

2) Similarly, I would suggest that your 2nd hypothesis needs to be more clear. What specific markers of physiologic derangement do you consider clinically meaningful here? Please revise the terminology in your hypothesis.

3) I would argue that for each hypothesis you should have a primary endpoint; so on p7, lines 148-152, I would identify one or a couple parameters as your primary endpoint for your 2nd hypothesis. Then, for secondary outcomes, you should be more specific about elements like "indices of cardiovascular and cardiopulmonary function." For example, did you measure cardiac output in some manner? Or vasopressor use? Or ventilator settings?

4) In your conclusions, I disagree with your statement that "XSTAT 30 pellets can be easily identified and removed prior to definitive abdominal closure...." In my experience with these pellets, once they get saturated with blood, they are actually challenging to find, consistent with your finding that pellets were left behind in 2 animals. You also did not study definitive closure (I would not consider necropsy the same as definitive closure); so please revise this comment accordingly.

Minor issues:

1) I would suggest mentioning the concept of intra-abdominal foam in the introduction

2) The background of your abstract is a bit awkward. For example, "co-morbidities" should probably be better described as "adverse outcomes" related to prolonged ischemia and subsequent reperfusion.

and then detail your specific endpoints in your methods section.

3) Can you provide more detail on your randomization approach?

4) Methods, p.4. The protocol number is unnecessary. Lines 83-91 can be eliminated by referencing your prior work.

5) Please specify the total (or approximate total) of pellets injected in the FP animals and then list a total number of pellets per animal in FP and BP in either the narrative or one of your tables.

6) Results, p.9. I would not consider a 4 second difference clinically meaningful. Consider measuring this in minutes which would show no difference.

7) Results, p.11. Can you indicate the number of animals and the group assignments for the TI ischemic changes? Was this due to hypoperfusion?

8) Figure 3. Legend. Do you mean "respective"? Did you also compare FP and BP at each timepoint?

Thank you again for the opportunity to review this interesting study.

7. PLOS authors have the option to publish the peer review history of their article (what does this mean?). If published, this will include your full peer review and any attached files.

Reviewer #1: No

Reviewer #3: **Yes: **Jeremy W. Cannon, MD, SM

---

## [Author Response · Author response to Decision Letter 3]

20 Oct 2020

Author Responses to Reviewers’ Comment and Questions

Reviewer #1: The authors have persisted in comparing their approach to reboa, yet have provided inadequate balance in their presentation. This is unfortunate, as comparing their experimental data to reboa is not needed in this precclinical paper. Lastly, when pacing a reboa, either in the ED or prehospital, it is unknown if an aortic injury is present.

Authors’ Response: The authors continue to discuss the points raised by the reviewer at great length and have enjoyed the collegial academic debate. It is possible, at this stage, that an academic impasse has been reached. There is an endovascular approach to the management of NCTH and there are approaches that are not endovascular; XSTAT & Foam. While foam technology is a better analogue for comparison, ResQ foam is no longer considered a viable option by the US DoD and USASOC and therefore not an appropriate technological context for XSTAT. The authors would suggest that REBOA is an important “contrast” as opposed to a comparison with XSTAT. Known or suspected aortic injury remains a critical contraindication to placement of a REBOA catheter and in the authors’ opinions exemplifies why REBOA is an important technology that provides contrast to the XSTAT for management of NCTH. We would again like to thank Reviewer 1 for his/her critical input and response to our work.

Reviewer #3: Thank you for the opportunity to review this work by Bonanno and colleagues on a novel application of hemostatic pellets in an uncontrolled torso hemorrhage swine model. The authors have conducted a rigorous evaluation of this novel approach and have some very interesting results. I have a few comments and questions for the authors that I hope they will address.

Authors’ Response: The authors would like to thank Dr. Cannon for his thorough review of their manuscript and have addressed his comments in the following:

RQ1: Although I think I understand the meaning of the term "dyshomeostasis" I am not certain this is a very common term and would suggest that it is not specific enough to be considered an endpoint for your 2nd hypothesis. I would suggest using a different term to convey your meaning.

AR1: The authors have changed the wording of the abstract and the manuscript to appeal to a broader readership including the replacement of the word “dyshomeostasis” with “derangement(s)”. Additionally, the authors recognize that the second hypothesis was incompletely expressed and have revised it significantly to reflect specific metabolic parameters as evidence of the null hypothesis. 

RQ2: Similarly, I would suggest that your 2nd hypothesis needs to be more clear. What specific markers of physiologic derangement do you consider clinically meaningful here? Please revise the terminology in your hypothesis.

AR2: The authors concur and have revised accordingly (see above).

RQ3: I would argue that for each hypothesis you should have a primary endpoint; so on p7, lines 148-152, I would identify one or a couple parameters as your primary endpoint for your 2nd hypothesis. Then, for secondary outcomes, you should be more specific about elements like "indices of cardiovascular and cardiopulmonary function." For example, did you measure cardiac output in some manner? Or vasopressor use? Or ventilator settings?

AR3: The authors concur with the recommendation for organization and clarity of endpoints for each hypothesis and have revised accordingly. All endpoints (primary and secondary for each hypothesis) were recorded and analyzed and have been indicated as such in the specified paragraph.

RQ4: In your conclusions, I disagree with your statement that "XSTAT 30 pellets can be easily identified and removed prior to definitive abdominal closure...." In my experience with these pellets, once they get saturated with blood, they are actually challenging to find, consistent with your finding that pellets were left behind in 2 animals. You also did not study definitive closure (I would not consider necropsy the same as definitive closure); so please revise this comment accordingly.

AR4: The authors recognize that there is a difference between clinical application and controlled research regarding the performance of medical devices. Reasonably, the authors can only comment on direct observations from the model and this is what the original text indicated. However, the authors concur that the conclusions do not necessarily reflect translational research outcomes and encroach upon the clinical experience. In light of that recognition, the authors have significantly revised the conclusions to reflect opinions based on the research setting.

Minor issues:

RC1: I would suggest mentioning the concept of intra-abdominal foam in the introduction

At the time of the initial draft of this manuscript the authors considered including discussion of the prominent intra-abdominal foam device in the background section of the manuscript. As our laboratory was directly involved in USASOC validation testing of ResQ foam devices prior to the experimentation with XSTAT we felt that the discussion of intra-abdominal foam was not relevant to this manuscript. At that time, SOCOM had expressed strong disinterest in the foam device and an intent to discontinue support for its development. As it would not be pursued by the US DoD for further use in military casualty care we thought it irrelevant as a contextual background for XSTAT. Since the submission of this manuscript, the US DoD has invested heavily in foam technology with a new industry partner. Unfortunately, at this time we are unable to discuss that work as it is proprietary to the industry partner and publication of foam translational data has not occurred in the public space. Future publications regarding XSTAT for NCTH will certainly discuss alternative NCTH technologies as circumstances allow.

RC2: The background of your abstract is a bit awkward. For example, "co-morbidities" should probably be better described as "adverse outcomes" related to prolonged ischemia and subsequent reperfusion.

and then detail your specific endpoints in your methods section.

AR2: The authors concur and have revised the abstract accordingly. The second partial sentence in this comment is unclear and therefore has not been addressed.

RC3: Can you provide more detail on your randomization approach?

AR3: The manuscript has been revised to include randomization method.

RC4: Methods, p.4. The protocol number is unnecessary. Lines 83-91 can be eliminated by referencing your prior work.

AR4: The authors have removed the protocol number as suggested. After some deliberation, the authors chose to leave the methods section referenced by the reviewer in the manuscript for completeness as they are critical to the reader understanding how physiologic parameters presented in the results were recorded.

RC5: Please specify the total (or approximate total) of pellets injected in the FP animals and then list a total number of pellets per animal in FP and BP in either the narrative or one of your tables.

AR5: This is an excellent catch by the reviewer. The authors have clarified pellet number total administration in the text of the methods. 

RC6: Results, p.9. I would not consider a 4 second difference clinically meaningful. Consider measuring this in minutes which would show no difference.

AR6: The authors recognize and agree that a four second difference is not meaningful at the level of definitive care. However, four seconds is highly relevant to a medic providing “care under fire”. This parameter was discussed directly with USASOC CDD during the design of the study and therefore has been retained as written. 

RC7: Results, p.11. Can you indicate the number of animals and the group assignments for the TI ischemic changes? Was this due to hypoperfusion?

AR7: These data were indicated in the manuscript and are found on P12 line 233 of the revised manuscript. The authors did not have the appropriate data to speculate the source of the ischemic changes.

RC8: Figure 3. Legend. Do you mean "respective"? Did you also compare FP and BP at each timepoint?

AR8: The authors have updated the Figure 3. Legend appropriately. Yes, the authors compared FP and BP at each time point though no significant differences were detected between groups.

---

## [Editor Report · Decision Letter 4]

23 Oct 2020

Novel Use of XSTAT 30™ for Mitigation of Lethal Non-Compressible Torso Hemorrhage in Swine

PONE-D-19-14707R4

Dear Dr. Ross,

We’re pleased to inform you that your manuscript has been judged scientifically suitable for publication and will be formally accepted for publication once it meets all outstanding technical requirements.

Kind regards,

Erin Lavik

Academic Editor

PLOS ONE
---

## [Editor Report · Acceptance letter]

6 Nov 2020

PONE-D-19-14707R4 

Novel Use of XSTAT 30 for Mitigation of Lethal Non-Compressible Torso Hemorrhage in Swine 

Dear Dr. Ross:

I'm pleased to inform you that your manuscript has been deemed suitable for publication in PLOS ONE. Congratulations! Your manuscript is now with our production department. 

Kind regards, 

on behalf of

Dr. Erin Lavik 

Academic Editor

PLOS ONE